# Genetic Characterization of Native Donkey (*Equus asinus*) Populations of Turkey Using Microsatellite Markers

**DOI:** 10.3390/ani10061093

**Published:** 2020-06-24

**Authors:** Selen Yatkın, Fulya Özdil, Emel Özkan Ünal, Serdar Genç, Selçuk Kaplan, Eser Kemal Gürcan, Sezen Arat, Mehmet İhsan Soysal

**Affiliations:** 1Department of Agricultural Biotechnology, Tekirdağ Namık Kemal University, 59030 Tekirdağ, Turkey; selenyatkin@nku.edu.tr (S.Y.); sarat@nku.edu.tr (S.A.); 2Department of Animal Science, Tekirdağ Namık Kemal University, 59030 Tekirdağ, Turkey; egurcan@nku.edu.tr (E.K.G.); misoysal@nku.edu.tr (M.İ.S.); 3Department of Agricultural Biotechnology, Kırşehir Ahi Evran University, 40100 Kırşehir, Turkey; serdar.genc@ahievran.edu.tr; 4Faculty of Veterinary Medicine, Tekirdağ Namık Kemal University, 59030 Tekirdağ, Turkey; skaplan@nku.edu.tr

**Keywords:** *Equus asinus*, donkey, genetic diversity, molecular markers, microsatellites, Turkey

## Abstract

**Simple Summary:**

This study was conducted to evaluate the genetic variability of Turkish native donkey (*Equus asinus*) populations, using polymorphism of 17 microsatellite markers. The results revealed a highly mixed genotype of all the examined donkeys, suggesting that two different group of breeds can be distinguished from each other on the basis of microsatellite markers.

**Abstract:**

This study presents the first insights to the genetic diversity and structure of the Turkish donkey populations. The primary objectives were to detect the main structural features of Turkish donkeys by microsatellite markers. A panel of 17 microsatellite markers was applied for genotyping 314 donkeys from 16 locations of Turkey. One hundred and forty-two alleles were identified and the number of alleles per locus ranged from 4 to 12. The highest number of alleles was observed in AHT05 (12) and the lowest in ASB02 and HTG06 (4), while ASB17 was monomorphic. The mean *H_O_* in the Turkish donkey was estimated to be 0.677, while mean *H_E_* was 0.675. The polymorphic information content (PIC) was calculated for each locus and ranged from 0.36 (locus ASB02) to 0.98 (locus AHT05), which has the highest number of alleles per locus in the present study. The average PIC in our populations was 0.696. The average coefficient of gene differentiation (G_ST_) over the 17 loci was 0.020 ± 0.037 (*p* < 0.01). The G_ST_ values for single loci ranged from −0.004 for LEX54 to 0.162 for COR082. Nei’s gene diversity index (H_t_) for loci ranged from 0.445 (ASB02) to 0.890 (AHT05), with an average of 0.696. A Bayesian clustering method, the Structure software, was used for clustering algorithms of multi-locus genotypes to identify the population structure and the pattern of admixture within the populations. When the number of ancestral populations varied from K = 1 to 20, the largest change in the log of the likelihood function (ΔK) was when K = 2. The results for K = 2 indicate a clear separation between Clade I (KIR, CAT, KAR, MAR, SAN) and Clade II (MAL, MER, TOK, KAS, KUT, KON, ISP, ANT, MUG, AYD and KAH) populations.

## 1. Introduction

Turkey is one of the important bio-geographical countries which encompass three important biodiversity hotspots such as the Caucasus, Iran-Anatolian, and Mediterranean basin [1]. Also Turkey has quite a wide range of biodiversity on account of geomorphologic, topographic features, the variety of climate and geographic conditions, either flora and fauna. This biodiversity has included a high amount of endemic and rare species [2].

Numerous factors such as technological improvement, population explosion, industrialization are responsible for the disturbance of natural balance permanently. For this reason, in order to conserve the plant and animal biodiversity, especially against the threat of endanger or disappearance, protection and conservation strategies have to be formed. The morphological and genetic identification of the species for the establishment of these strategies become crucial issues for the conservation of biodiversity. Without these protective strategies, significant changes are likely to occur in the ecosystem balance and biodiversity level.

Turkey is an agricultural country where plants and animals have been raised since ancient times. For this reason, it is considered that many local animal breeds were first bred here and spread to other geographical regions of the world. Turkey has rich animal genetic resources and donkeys are one of these resources, but yet so little information is verified for the donkey breeds of Turkey. Donkey breeds that helped the breeders under severe natural conditions, and have assisted in transportation for centuries, and are still used in some rural areas in Turkey, are under threat of extinction due to the impact of industrialization [3,4,5]. In the last years, donkey populations have declined dramatically in Turkey. For these reasons, the conservation strategies have to be explored and breeds must be identified before the loss of genetic resources.

To date, little information has been found about the donkey breeds of Turkey; even some of the donkey breeds of Turkey had been extinct without clear identification. According to the Food and Agriculture Organization of the United Nations (FAO), Domestic Animal Diversity Information System (DAD-IS), Turkey has three native donkey breeds: Anatolian, Merzifon and Karakaçan donkey breeds. But on the other hand, in other unpublished records, Mardin White donkey (close to Mardin province), Toros donkey (close to Toros Mountains, Antalya province), Urfa Rahvan donkey (close to Şanlıurfa province) and Kars Yorga donkey (close to Kars Province that is located in extremely north east part of Turkey) breeds are also found in Turkey [6].

So the aim of this study has threefold: (i) to provide a detailed sampling across the entire country for molecular characterization to check if the above mentioned breeds are still found in Turkey or not, (ii) to analyze the genetic diversity of donkeys raised in Turkey using a set of 17 microsatellite markers, (iii) and to determine the genetic relationship and characterize geographical and genetic differentiation between different donkey breeds at different sites in Turkey. This research is the first application of molecular markers to characterize the donkey breeds raised in Turkey.

## 2. Materials and Methods

### 2.1. Sampling and DNA Isolation

According to the FAO database (Domestic Animal Diversity Information System, accessed Sep 15, 2019); the estimated number of donkeys in Turkey is about 150,000. In this study, a total of 314 blood samples from different individuals were collected from 16 different locations (Figure 1). These locations were selected to represent the expected native Turkish donkey breeds: Anatolian, Merzifon, Karakaçan etc. The locations and the potential distribution of these breeds with the number of sample sizes are given in Table 1.

Blood samples were taken to Ethylenediaminetetra-acetic acid (EDTA) (0.5 mM, pH 8.0) coated vacutainer tubes and stored at +4 °C until DNA extraction. Approximately 10 mL of blood per animal was collected and genomic DNA was extracted from whole blood using the standard phenol-chloroform-isoamyl alcohol (25:24:1) extraction method. Extracted DNA was diluted in TE buffer (10:1) and stored +4 °C till analysis.

### 2.2. Microsatellite Genotyping

In this study, twenty microsatellite markers (AHT05, ASB02, ASB17, ASB23, COR007, COR018, COR022, COR058, COR071, COR082, HMS02, HMS03, HMS07, HMS20, HTG06, HTG07, HTG10, LEX54, LEX73 and VHL209) which have been widely used and recommended for individual identification and parentage verification of equines were selected according to FAO’s guidelines [7]. Seventeen out of the 20 tested loci yielded clear PCR products and used for the analysis. ASB17, COR022 and LEX73 were excluded from the study. Genomic characteristics of the 17 microsatellite loci, primer sequences, fluorescent labels, allele size range, annealing temperatures, repeat motifs and the references are given in Table 2.

All the microsatellite loci were amplified by Polymerase Chain Reaction (PCR) and each forward primer was labeled with fluorescent dyes (6-FAM^TM^, VIC®, NED^TM^, PET®) at the 5’ end. Four multiplex PCRs using fluorescently labeled primers were developed (1st 4plex: HMS07, ASB23, HMS02, COR058; 2nd 5plex: HMS03, VHL209, ASB02, HMS20, COR007; 3rd 4plex: HTG07, AHT05, LEX54 and HTG10; 4th 4plex: HTG06, COR018, COR071, COR082). The reactions were performed in a total 10 µL volume. PCRs were comprised with approximately 50 ng DNA, 1X PCR buffer 1.5 mM MgCl_2_, 0.2 mM dNTPs, 0.050 pmol of each primers, 1 U Taq DNA Polymerase (Invitrogen Taq DNA Polymerase).

The cycling conditions included an initial denaturation step at 95 °C for 5 min, 35 cycles of 95 °C for 1 min, annealing temperature between 55–62 °C (see Table 2) for 45–60 s, elongation step at 72 °C for 1 min and a final extension at 72 °C for 5 min. Amplification was carried out using a Veriti™ 96–Well Thermal Cycler or ProFlex PCR System (Applied Biosystems, Foster City, CA, USA).

PCR products were checked in 2% agarose gel stained with SYBR™ Safe DNA Gel Stain. The samples were mixed with formamide and LIZ® 500-bp internal size standard (Applied Biosystems™) and detected by capillary electrophoresis using a 3500 XL Genetic Analyzer® (Applied Biosystems™) sequencer. Allele sizes were determined with the *GeneMapper*® Software V4.0 (Applied Biosystems™).

### 2.3. Statistical Analysis

Genetic diversity parameters were estimated for each microsatellite locus and across all loci for each population by total number of alleles, the mean number of alleles (Na), effective number of alleles (Ne), polymorphic information content for each locus (PIC), observed heterozygosity (H_O_), expected heterozygosity (H_E_), private alleles (N_P_) Hardy–Weinberg equilibrium and null allele frequencies using Genetix v4.05 [21], FSTAT v2.9.4 [22], POPGENE Version 1.31 [23] and GenAlEx Version 6.5 [24]. Wright’s F statistics (F_ST_, F_IS_ and F_IT_) as proposed by Weir and Cockerham [25] were computed using Genetix® software. Nei’s gene diversity (H_T_), diversity between breeds (D_ST_) and coefficient of gene differentiation (G_ST_) values were calculated with FSTAT v2.9.4 [22]. Exact tests for deviation from the Hardy-Weinberg (HW) equilibrium and partitioning of genetic diversity using analysis of molecular variance (AMOVA) were performed using the ARLEQUIN v. 3.5.2.2 [26]. Pairwise genetic distances (Reynold’s genetic distance) and Nei’s [27] unbiased D_A_ genetic distances were calculated using the Populations v 1.2.30 software. Neighbour-net dendrogram constructed from Reynold’s genetic distances by using SplitsTree v4.16.0 [28].

The genetic structure of the populations was investigated using STRUCTURE 2.3.4, (Oxford, UK) [29,30,31]. Analysis was performed with a burn of 500,000 in length, followed by 500,000 Markov chain Monte Carlo iterations for each from K= 1–18, with 20 replicate runs for each K, using independent allele frequencies and an admixture model. Evanno’s method [32] was used to identify the appropriate number of clusters using ΔK, based on the rate of change in the log probability of the data. The optimal K values were selected by means of STRUCTURE HARVESTER [33]. This software, a web-based program, was used for collating the results generated by the program STRUCTURE. The clustering pattern was implemented in the CLUMPP program and visualized using the software DISTRUCT software version 1.1 [34].

## 3. Results

Out of the 20 microsatellites analyzed, the following 17 showed useful data in study population of the Turkish donkey: AHT05, ASB02, ASB23, COR007, COR018, COR058, COR071, COR082, HMS02, HMS03, HMS07, HMS20, HTG06, HTG07, HTG10, LEX54 and VHL209 (Table 2). The ASB17 COR022 and LEX73 loci were excluded from the analysis. The COR022 and LEX73 loci failed to amplify in all samples and did not provide data for reliable analysis while the ASB17 marker was monomorphic (91 bp) for all examined animals.

In this study, the mean number of alleles (N_a_), the number of effective alleles (N_e_), the number of private alleles (N_p_), expected (H_E_) and observed (H_O_) heterozygosity, F_IS,_ and PIC values for each population were given in Table 3. Among 17 tested loci, the number of alleles varied from 4 (ASB02 and HTG06) to 12 (AHT5), while the total number of observed alleles was 142 with an average of 8.235 per locus in 314 examined donkeys. The mean number of alleles observed in populations differed slightly: the minimum 4.529 was observed in AYD (AER) and the maximum 6.706 was in KIR (MAR). The observed (*H_O_*) and unbiased expected (*H_E_)* heterozygosities per location ranged from 0.6266 (KIR) to 0.7139 (AYD) and 0.6294 (ISP) to 0.6983 (ANT), respectively. The mean *H_O_* in the Turkish donkey was estimated to be 0.677, while mean *H_E_* was 0.675 (Table 3). The mean number of alleles N_a_ was the highest in KIR (6.706) and lowest in AYD donkeys (4.529). F_IS_ value within the populations varied between −0.0557 in KAS and 0.0923 in KIR population, wheras F_IS_ was statistically significant only for KIR breeds due to the deficiency of heterozygosity. A total of 13 private alleles were identified in the present work, and most of the private alleles (ten) were at low frequencies of below 5%. Three alleles unique to KAH (0.115), MAL (0.050) and AYD (0.083) showed a frequency that exceeded 5% (Table 3).

The characteristics of the analyzed loci along with the genetic variability statistics were summarized in Table 4. The total number of alleles per locus ranged from 4 (ASB02, HTG06) to 12 (ATH005), while the mean number of alleles per locus varied between 3.938 and 9.625 for the same loci, with a mean number of alleles per locus of 5.714. The number of effective alleles per locus (Ne) indicates the genetic diversity. Ne varied between 2.154 (ASB02) and 6.966 (ATH05), with a mean of 4.40 ± 2.21. For all the analyzed samples, the Na is higher than Ne, which indicates a relatively high genetic diversity. The polymorphic information content (PIC) was calculated for each locus and ranged from 0.36 (locus ASB02) to 0.98 (locus AHT05), which has the highest number of alleles per locus in the present study. The average PIC in our populations was 0.696. Thirteen microsatellites (ASB23, HMS02, COR058, HMS03, HMS20, COR007, HTG07, COR018, COR071, COR082, HTG06, LEX54 and AHT05), having a PIC value higher than the threshold of (0.5). Additionally, four loci (HMS07, VHL209, ASB02 and HTG10) showed moderate polymorphism (PIC > 0.25) (Table 4). The observed heterozygosity (H_O_) per locus was 0.418 (COR082) to 0.857 (AHT05), 0.658 on average. The expected heterozygosity per locus was 0.438 (HMS07) to 0.884 (AHT05), 0.670 on average. Mean H_O_ and H_E_ were higher than 0.418 for all loci. However, the value of H_O_ for 7 loci (ASB23, COR058, HMS03, ASB02, HTG07, COR082 and AHT05) was lower than the value of H_E_, indicating an excess of homozygosity. The highest F_IS_ value was observed in marker COR082 (0.2959), while the lowest F_IS_ was recorded for locus HMS07 (−0.0602). The maximum and minimum F_IT_ values were found in markers COR082 and HMS07, respectively. F_ST_ values ranged from 0.0000 to 0.1955, and the average values of F_IS_, F_ST_ and F_IT_ were 0.0194, 0.0192 and 0.0382 accordingly. Mean F_ST_ (0.0192) was moderate to low while H_S_ (0.683) was relatively high. Obtained overall D_ST_ value describing the diversity between breeds was 0.014. The average coefficient of gene differentiation (G_ST_) over the 17 loci was 0.020 ± 0.037 (*p* < 0.01). The G_ST_ values for single loci ranged from −0.004 for LEX54 to 0.162 for COR082. Nei’s gene diversity index (H_t_) for loci ranged from 0.445 (ASB02) to 0.890 (AHT05), with an average of 0.696. The presence of null alleles, defined as non-amplifying alleles, due to mutations at PCR priming sites, causes overestimation of both F_ST_ and genetic distance values. The null allele frequencies ranged from 0.000 (VHL209, ASB02) to 0.1894 (HTG06). The null allele frequencies in the studied microsatellite loci were below 20% (Table 4). The lowest and highest null allele frequencies were 0.000 (VHL209, ASB02) and 0.1894 (HTG06), respectively.

The interbreed genetic distance, or F_ST_ values of pairwise comparisons among the Turkish donkey populations are shown in Figure 2. In some cases, negative values were observed, and these equate to zero F_ST_ values. Our F_ST_ values fall into a small range, 0.00–0.056 (varying from white, and light to dark blue colors in Figure 2).

The neighbour-net phylogeny drawn from Reynold’s genetic distances (Figure 3) visualizes the relationships between Turkish donkey populations. Populations that shared close genetic relationships were placed on different branches that originated from the same basal node. It identifies four distinct clusters, which are clearly separated, i.e., (I) from the SAR-EAR-MRM region donkey populations, (II) and (III) from the AER-MRD region donkey populations and (IV) from the MRM, BSR, MDR, CAR and AER region donkey populations in Turkey (Figure 3). The phylogeny of Reynold’s distances was similar to that generated using the Nei’s D_A_ distances (Appendix A).

The analysis of molecular variance (AMOVA) is a useful tool to check how the genetic diversity is distributed within and among populations, whose structure is quantified by F_ST_. We analyzed different possible structures by creating and comparing different population groups. We ran the analysis under two hypotheses:

Hypothesis (1) seven groups according to the geographical distribution, i.e., group 1: The Marmara region (MAR) populations (KIR, CAT, MAL); group 2: The Black Sea region (BSR) populations (MER, TOK, KAS); group 3: The Aegean region (AER) populations (KUT, MUG, AYD); group 4: The Central Anatolia region (CAR) population (KON); group 5: The Mediterranean region (MDR) populations (ISP, KAH, ANT), Eastern Anatolia region (EAR) population (KAR) and group 7: South East Anatolian region (SAR) populations (MAR, SAN).

Hypothesis (2) four groups according to the Reynold’s genetic distances distribution, i.e., group 1: MRM region (KIR, CAT), SAR region (SAN, MA) and EAR region (KAR) populations; group 2: AER region (KUT) population and MDR region population (ISP); group 3: AER region population (AYD) and MDR region population (KAH); group 4: MRM region population (MAL), BSR region populations (MER, TOK, KAS), AER region population (MUG), CAR region population (KON) and MDR region population (ANT).

The Table 5 reports the results for the AMOVA analysis according to two hypotheses. The results highlight that most of the observed variance is due to differences within individuals. The Hypothesis (1) AMOVA (Table 5) analyses results showed that the variation among groups, among populations within groups, among individuals within populations, and within individuals were 1.07%, 0.96%, 1.69% and 96.29%, respectively. Variance components among groups and among individuals within populations were significant (*p* < 0.001) for all the studied loci (Table 5), demonstrating significant geographical distribution in studied donkey populations. Furthermore, variance component among populations within groups and within individuals were significant (*p* < 0.05). The Hypothesis (2) AMOVA analyses results showed that the variation among groups, among populations within groups, among individuals within populations, and within individuals were 2.16%, 0.49%, 1.68% and 95.67%, respectively. Variance components among groups, among individuals within populations and within individuals were significant (*p* < 0.001) for all the studied loci demonstrating significant Reynold’s genetic distances distribution in studied donkey populations (Table 5). Furthermore, variance component among populations within groups were significant (*p* < 0.05).

The populations’ structure (Figure 4) was analyzed using Bayesian clustering analysis to determine the number of clusters (K) present in the populations, permitting the identification of differences among populations and hidden substructures within them. The genetic structure of each population was determined based on admixture level for each donkey individual using correlated allele frequencies model implemented within the STRUCTURE software. When the number of ancestral populations varied from K = 1 to 20, the largest change in the log of the likelihood function (ΔK) was when K = 2 (Figure 4). The results for K = 2 (Figure 4) indicate a clear separation between Clade I (KIR, CAT, KAR, MAR, SAN) and Clade II (MAL, MER, TOK, KAS, KUT, KON, ISP, ANT, MUG, AYD and KAH) populations. In Figure 4, each individual is represented by a single vertical line broken into K colored segments. The mixed colors with proportional lengths represent the admixture level for predefined populations of K2.

## 4. Discussion

In this study, 20 microsatellite markers (AHT05, ASB02, ASB17, ASB23, COR007, COR018, COR022, COR058, COR071, COR082, HMS02, HMS03, HMS07, HMS20, HTG06, HTG07, HTG10, LEX54, LEX73 and VHL209) were used for genetic characterizing the population genetic structure and genetic diversity of 16 donkey population that cover the 7 geographical regions in Turkey. The ASB17, COR022 and LEX73 loci were excluded from the statistical analysis. The COR022 and LEX73 loci failed to amplify in all the samples and did not provide data for reliable analysis while the ASB17 marker was monomorphic (91 bp) for all examined animals. Similar results were reported in the Balkan donkey populations [35]. This study is the first systematic large-scale study of different geographical regions populations from Turkey by using the microsatellites.

The 17 microsatellite analysis revealed relatively high level of genetic diversity between individuals, but no high significant differences in the main genetic characteristics of the geographical region’s population (MRM, BSR, AER, CAR, MDR, EAR, SAR): number of alleles (N_a_), effective number of alleles (N_e_), and expected and observed heterozygosities. These results are comparable to the previous values reported in Balkan donkeys [35], Spanish donkeys [36] and Catalonian donkeys [37]. In terms of mean number of alleles, the genetic variability observed in the 16 Turkish donkey populations was lower than that reported in Catalonian donkey breed [37], five Spanish breeds [36], three Croatian donkey populations [38], 15 indigenous Chinese donkey breeds [39,40,41,42], Balkan donkeys [35] and Tunisian donkeys [43], but higher than that observed in the seven indigenous Italian donkey breeds [44,45,46,47].

The results of microsatellite polymorphism revealed relatively high degree of heterozygosity in the Turkish donkey populations investigated in this study. Among Turkish donkey populations, the H_E_ ranged from 0.6294 (ISP) to 0.6983 (ANT), which showed a comparable level to the previous values reported in Spanish donkeys [36], Catalonian donkey breeds [37], Croatian coast donkey populations [48], Balkan donkey breeds [35], Tunisian donkeys [43], Chinese donkey breeds [39,40,41,42] and was more diversified than Italian [44,45,46,47] and American donkeys [49]. This finding indicates that there are appreciable differences in the level of genetic variability among 16 Turkish donkey populations.

F_IS_ index indicates the excess of homozygosity in a subpopulation and, with reference to molecular markers, informs if a pattern of reduction in diversity owing to several causes exists. In this study, F_IS_ ranged from a minimum of −0.0557 in KAS to a maximum of 0.0923 in KIR population, whereas F_IS_ was statistically significant only for KIR breeds due to the deficiency of heterozygosity. The highly significant (*p* < 0.001) F_IS_ value (0.0923) revealed a rather high inbreeding degree within populations. The heterozygote deficiency found in the KIR population, could be due to the higher rate of inbreeding, to the population subdivision (Wahlund effect), and to the presence of “null alleles” (non-amplifying alleles).

Null alleles, defined as nonamplifiable alleles due to mutations in the PCR binding site, cause only a single allele to peak like a homozygote, thus cause erroneous readings. These alleles cause overestimation of both F_ST_ and genetic distance values. It was reported by Dakin and Avise [50] that null allele frequencies below 0.20 have no significant effect on paternity tests. When the null allele frequencies obtained are examined, it is seen that the null allele frequency values of 17 microsatellites to be studied are below 0.20. The lowest and highest null allele frequencies were 0.000 (VHL209, ASB02) and 0.1894 (HTG06), respectively. Taking this value into consideration, it has been demonstrated that the studied loci can be safely used in paternity tests.

Private alleles are alleles that are found only in a single population among a broader collection of populations. These alleles have proven to be informative for diverse types of population-genetic studies. These private alleles are present in greater numbers in differentiated donkey breeds [36,38,42]. There were 13 private alleles among Turkish donkey breeds and most of the private alleles (ten) were at low frequencies of below 5%. Three alleles unique to KAH (0.115), MAL (0.050) and AYD (0.083) showed a frequency that exceeded 5%. In the present study, these alleles were consistent with those found by other authors [36,38,42], although we observed a higher value in the province of KIR and KAR, which is genetically similar to these two populations than the other provinces. Most of the KIR population (60%), samples are collected from the donkey farm in Kırklareli. In this farm, the individuals were collected from Eastern and South Eastern Anatolian region of Turkey.

PIC is a parameter indicative of the degree of informativeness of a marker. The PIC value may range from 0 to 1. In the studied Turkish donkey population, the average PIC value was 0.696 ranging from 0.3600 (ASB02) to 0.9800 (AHT05). When PIC > 0.5, 0.5 > PIC > 0.25, and PIC < 0.25, it indicates the locus has high polymorphism, moderate polymorphic, and low polymorphism, respectively [51]. Thirteen microsatellites (ASB23, HMS02, COR058, HMS03, HMS20, COR007, HTG07, COR018, COR071, COR082, HTG06, LEX54 and AHT05), having a PIC value higher than the threshold of 0.5 [51,52], seemed to be highly informative and can be used in quantifying the genetic diversity and also in paternity studies in Turkish donkey population. Additionally, four loci (HMS07, VHL209, ASB02 and HTG10) showed moderate polymorphism (PIC > 0.25). PIC values calculated in the present investigation were comparable with those reported by Aranguren – Mèndez et al. [36] in 5 endangered Spanish donkey breeds (0.20–0.85), Ivankovic et al. [38] in 3 Croatian donkey populations (0.36–0.78), Bordonaro et al. [46] in Pantesco and two Sicilian autochthonous donkey breeds (0.146–0.796), Matassino et al. [47] in two Italian autochthonous donkey breeds (0.1918–0.8522), Zhang et al. [40] in 10 Chinese donkey breeds (0.7218–0.7967), Stanisic et al. [35] in Balkan donkey breeds (0.07–0.84) and by Zeng et al. [42] in Chinese donkey breeds (0.1489–0.8670). The variability in PIC values found in literature may be due to different microsatellite markers used in the studied populations. In the present study, the high PIC values prove that the microsatellite markers used are highly polymorphic and can be well utilized for studying the genetic diversity in Turkish donkey populations. 

In this study, the mean value of between-population diversity value (D_ST_), coefficient of gene diversity (G_ST_) and Nei gene diversity (H_T_), were determined as 0.014, 0.020, and 0.696, respectively. The global mean of the genetic diversity value (D_ST_) indicated that the low diversity among 16 Turkish donkey populations studied. Nei’s gene diversity values (H_T_) was considerably lower than Zhang et al. [40] in Chinese donkey breeds, but similar to Aranguren—Mèndez et al. [36] in 5 endangered Spanish donkey breeds. The average G_ST_ value obtained from overall loci pointed out that 2% of total genetic variation resulted from the differences between the populations. In all other respects, it can be said that 98% genetic variation is caused by the difference between individuals. All studied loci showed a not significant deviation from the Hardy-Weinberg Equation. Null allele frequencies were lower than the reported value (20%) by Dakin and Avise [50]. These results indicated that the microsatellite markers studied may be safely used in genetic diversity studies in Turkish donkey populations. 

According to all the pairwise differences (Slatkins linearized F_ST_) in this study, the distribution of F_ST_ showed low genetic divergence (0.000 < F_ST_ < 0.05) among populations in general. The F_ST_ comparison values obtained were significant in 70 pairwise calculations (*p* < 0.05; *p* < 0.01; *p* < 0.001). The highest level of differentiation was observed between SAN–MAL, KAR–TOK and KUT–TOK populations (F_ST_ > 0.05) and the lowest between CAT–KIR, KON–MAR, KAS–MER, ANT–KAS and ANT–KON breeds (F_ST_ = 0.000), respectively. Neighbor-net representing the Reynolds distance confirmed these findings; Cluster II (KUT – ISP) and Cluster II (AYD – KAH) groups clustered in an intermediate position between the Cluster I (CAT, KIR, MAR, KAR, SAN) and Cluster IV (MER, MUG, KON, ANT, KAS, MAL, TOK). The F_ST_ value of Turkish donkey populations detected in this study was similar to the value of American donkeys [49], Chinese donkeys [42], but lower than that of donkeys in the Europe [45,46], Near East and northeast Africa [53].

An AMOVA was carried out to investigate the relative contribution of different factors to the observed genetic variability, with each factor considered in a separate analysis, i.e., seven groups according to the geographical prevalence (MAR, BSR, AER, CAR, MDR, EAR and SAR), four groups according to the Reynold’s genetic distances distribution (1st: MRM region (KIR, CAT), SAR region (SAN, MA) and EAR region (KAR) populations, 2nd: AER region (KUT) population and MDR region population (ISP); 3rd: AER region population (AYD) and MDR region population (KAH); 4rd group: MRM region population (MAL), BSR region populations (MER, TOK, KAS), AER region population (MUG), CAR region population (KON) and MDR region population (ANT)). AMOVA analysis results indicate that the majority of the observed variance is due to differences among individuals within populations. The most part of the variation is observed within the individuals (96.29% Hypothesis 1 and 95.67% Hypothesis 2) whereas the differences among groups represent only the 1.07% and 2.16% of the variation, respectively. These results are similar to a wide range of studies [38,45,54]. Among groups, among populations within groups, among individuals within populations were also a significant source of variation (*p* < 0.001, *p* < 0.05), although substantially smaller than the within individual’s component.

In this study the analysis with the STRUCTURE programme revealed that Turkish donkey populations were grouped into two lineages when K = 2 (Figure 4). Cluster I included SAN, KAR, MAR, KIR and CAT populations and Cluster II gathered MUG, ANT, KON, KAS, MAL, TOK and MER breeds, while other donkey populations (KUT, ISP, AYD and KAH) appeared to be the contact zone between both clusters, as individuals had mixed lineages. The STRUCTURE analysis results support the neighbour-net dendrogram results, as well as F_ST_ and genetic distance results. Our results provide a broad perspective on the extant genetic diversity and population structure of Turkish donkey populations. 

## 5. Conclusions 

Genetic diversity is the main component of the adaptive evolution mechanism because of its preeminent role for the long-term survival probability of all species. In summary, our results suggested the relatively high genetic diversity of 16 Turkish donkey populations and brought an insight in the structure of the analyzed populations. This study is the first attempt towards a comprehensive genetic characterization of Turkish donkey populations. Despite the decreasing population size, the genetic diversity of Turkish donkey population seems still conserved. Nevertheless, further studies should be conducted to deeply evaluate the genetic variability of the Turkish donkey breeds. Furthermore, the results of this study can be utilized for future breeding strategies and conservation.

## Figures and Tables

**Figure 1 animals-10-01093-f001:**
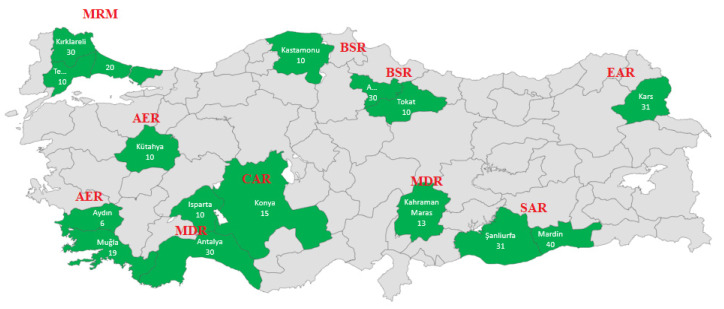
Geographic distribution of the 16 donkey populations included in the study. The details of the regions are given in Table 1.

**Figure 2 animals-10-01093-f002:**
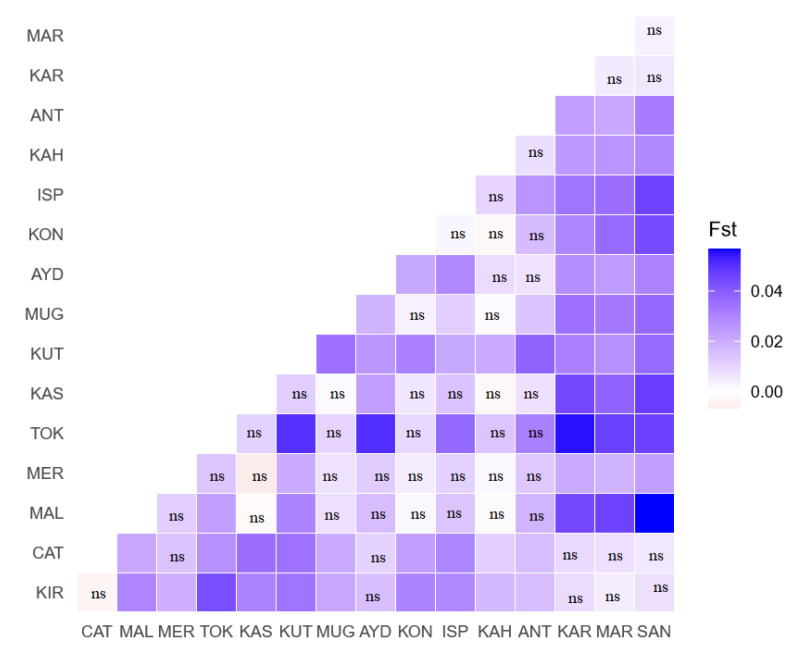
Graphical representation of pairwise F_ST_ distances between the 16 Turkish donkey populations studied. Color-codes are defined on the scale at the right side of the figure. (ns: not significant, blank significant *p* < 0.05; *p* < 0.01; *p* < 0.001). Kırklareli-KIR, İstanbul/Çatalca—CAT, Malkara—MAL, Amasya/Merzifon—MER, Tokat—TOK, Kastamonu—KAS, Kütahya—KUT, Muğla—MUG, Aydın—AYD, Isparta—ISP, Kahramanmaraş—KAH, Antalya—ANT, Konya—KON, Kars—KAR, Mardin—MAR, Şanlıurfa—SAN.

**Figure 3 animals-10-01093-f003:**
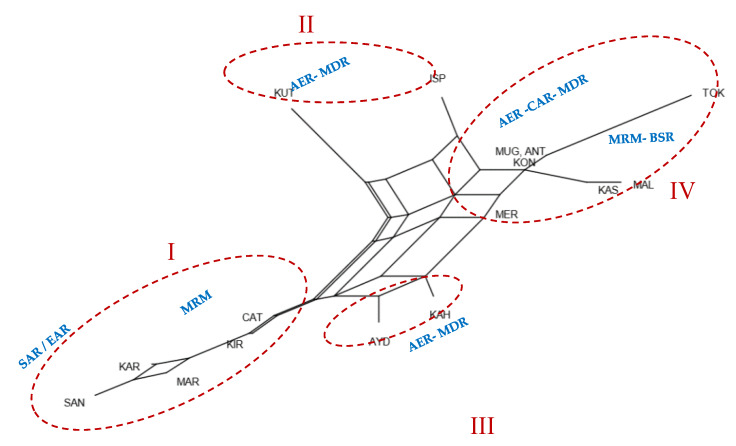
Neighbour-net dendrogram constructed from Reynold’s genetic distances among 16 Turkish donkey population (Kırklareli—KIR, İstanbul/Çatalca—CAT, Malkara—MAL, Amasya/Merzifon—MER, Tokat—TOK, Kastamonu—KAS, Kütahya—KUT, Muğla—MUG, Aydın—AYD, Isparta—ISP, Kahramanmaraş—KAH, Antalya—ANT, Konya—KON, Kars—KAR, Mardin—MAR, Şanlıurfa—SAN).

**Figure 4 animals-10-01093-f004:**
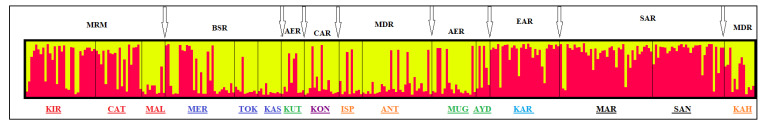
Clustering analysis by structure for the full-loci dataset assuming K = 2. Population name abbreviations are labeled below the structure result (Kırklareli—KIR, İstanbul/Çatalca—CAT, Malkara—MAL, Amasya/Merzifon—MER, Tokat—TOK, Kastamonu—KAS, Kütahya—KUT, Muğla—MUG, Aydın—AYD, Isparta—ISP, Kahramanmaraş—KAH, Antalya—ANT, Konya—KON, Kars—KAR, Mardin—MAR, Şanlıurfa—SAN). The geographical regions are labeled above the structure results (MRM: Marmara, BSR: Black sea region; AER: The Aegean region, CAR: The Central Anatolia region, MDR: The Mediterranean region, EAR: Eastern Anatolia region SAR: South East Anatolian region).

**Table 1 animals-10-01093-t001:** The regions, locations, the geographical locations with the number of sample sizes.

Region	Locations	Geographical Location		Number of Samples
**Marmara** **(MRM)**	Kırklareli	41°51′N	27°19′E	30
	İstanbul-Çatalca	41°06′N	28°30′E	20
	Tekirdağ-Malkara	40°52′N	26°57′E	10
**Black Sea** **(BSR)**	Amasya-Merzifon	40°53′N	35°32′E	30
	Tokat	40°12′N	36°27′E	10
	Kastamonu-Cide	41°50′N	32°54′E	10
**Aegean** **(AER)**	Kütahya	39°21′N	30°01′E	10
	Muğla	36°37′N	29°26′E	19
	Aydın	37°44′N	28°01′E	6
**Central Anatolia** **(CAR)**	Konya	37°38′N	32°26′E	15
**Mediterranean** **(MDR)**	Isparta	37°49′N	30°44′E	10
	Kahramanmaraş	37°30′N	36°57′E	13
	Antalya	36°50′N	30°13′E	30
**Eastern Anatolia** **(EAR)**	Kars	40°36′N	43°07′E	30
**South East Anatolia (SAR)**	Mardin	37°18′N	40°44′E	40
	Şanlıurfa	37°10′N	38°50′E	31
		**Total**		**314**

Marmara Region (MRM): Kırklareli—KIR, İstanbul/Çatalca—CAT, Malkara—MAL; Black Sea Region (BSR): Amasya/Merzifon—MER, Tokat—TOK, Kastamonu—KAS; Aegean Region (AER): Kütahya—KUT, Muğla—MUG, Aydın—AYD; Central Anatolian Region (CAR): Konya—KON; Mediterranean Region (MDR): Isparta—ISP, Kahramanmaraş—KAH, Antalya—ANT; Eastern Anatolia Region (EAR): Kars—KAR; South East Anatolian Region (SAR): Mardin—MAR, Şanlıurfa—SAN.

**Table 2 animals-10-01093-t002:** Genomic characteristics of the 17 microsatellite loci *; chromosome numbers, primer sequences with the fluorescent labels, annealing temperatures, GenBank accession numbers, allele size ranges, repeat motifs and the references.

Name(s)	Chromosome	Primer Sequence (5’→ 3’)	Annealing Temperature (°C)	GenBank Accession Number	Allele Range (bp)	Repeat Motif	Multiplex Group	Ref.
HMS07	1	F:FAM-CAGGAAACTCATGTTGATACCATC R: TGTTGTTGAAACATACCTTGACTGT	58	X74636	160–178	(AC)_2_(CA)n	1	[8]
ASB23	3	F: NED-GAGGTTTGTAATTGGAATG R: GAGAAGTCATTTTTAACACCT	58	X93537	128–154	(TG)_17_	1	[9]
HTG07	4	F: PET-CCTGAAGCAGAACATCCCTCCTTG R: ATAAAGTGTCTGGGCAGAGCTGCT	58	AF142607	272–297	(GT)n	3	[10]
AHT05	8	F: PET-ACGGACACATCCCTGCCTGC R: GCAGGCTAAGGGGGCTCAGC	58	-	130–146	(GT)n	3	[11]
HMS03	9	F:NED-CCAACTCTTTGTCACATAACAAGA R: CCATCCTCACTTTTTCACTTTGTT	58	X74632	150–170	(TG)2(CA)2TC(CA)n/(TG)2(CA)2TC(CA)Nga(CA)5	2	[8]
HMS02	10	F: NED-ACGGTGGCAACTGCCAAGGAAG R: CTTGCAGTCGAATGTGTATTAAATG	58	X74631	218–238	(CA)n(TC)_2_	1	[8]
COR058	12	F: VIC-GGGAAGGACGATGAGTGAC R: CACCAGGCTAAGTAGCCAAAG	56	AF108375	210–230	i(TG)23	1	[12]
VHL209	14	F: FAM-TCTTACATCCTTCCATTACAACTA R: TGATACATATGTACGTGAAAGGAT	56	Y08451	84–96	(AC)17	2	[13]
ASB02	15	F: FAM-CCTTCCGTAGTTTAAGCTTCTG R: CACAACTGAGTTCTCTGATAGG	54	X93516	222–254	(GT)24	2	[14]
HMS20	16	F: VIC-TGGGAGAGGTACCTGAAATGTAC R: GTTGCTATAAAAAATTGTCTCCCTAC	58	-	116–140	-	2	[15]
COR007	17	F: PET-GTGTTGGATGAAGCGAATGA R: GACTTGCCTGGCTTTGAGTC	56	AF083450	156–170	(GT)_18_	2	[16]
LEX54	18	F: FAM-TGCATGAGCCAATTCCTTAT R: TGGACAGATGACAGCAGTTC	55	AF075656	165–177	(AC)_18_	3	[17]
HTG06	15	F:FAM-CCTGCTTGGAGGCTGTGATAAGAT R: GTTCACTGAATGTCAAATTCTGCT	58	-	84–106	(TG)n	4	[18]
HTG10	21	F: VIC-CAATTCCCGCCCCACCCCCGGCA R: TTTTTATTCTGATCTGTCACATTT	54	AF169294	93–113	(TG)n/TATC(TG)n	3	[10]
COR018	25	F: FAM-AGTCTGGCAATATTGAGGATGT R: AGCAGCTACCCTTTGAATACTG	56	AF083461	249–271	İ(CA)_18_	4	[16]
COR071	26	F: PET-CTTGGGCTACAACAGGGAATA R: CTGCTATTTCAAACACTTGGA	56	AF142608	190–202	İ(TG)_17_/İ(AG)_18_	4	[19]
COR082	29	F: GCTTTTGTTTCTCAATCCTAGC R: TGAAGTCAAATCCCTGCTTC	58	AF154935	192–226	(AG)_n_	4	[20]

* ASB17, LEX73 and COR022 were excluded from the study.

**Table 3 animals-10-01093-t003:** Main diversity parameters from each Turkish donkey populations included in this study for a panel of 17 microsatellite markers: the number of individuals (N), the mean number of alleles (N_a_), the number of effective alleles (N_e_), the number of private alleles (N_p_), observed heterozygosity (H_o_), unbiased expected heterozygosity (H_e_) and deficit of heterozygotes (F_IS_).

Region ^1^	Location ^1^	N	N_a_	Ne	N_p_	H_O_	H_E_	F_ıs_
**MRM**	**KIR**	30	6.706	3.684	4	0.6266	0.6893	0.0923 ***
	**CAT**	20	6.294	3.712	-	0.6882	0.6946	0.0094
	**MAL**	10	5.118	3.274	1	0.6379	0.6806	0.0660
**BSR**	**MER**	30	6.294	3.618	-	0.6720	0.6851	0.0194
	**TOK**	10	5.000	3.207	-	0.6588	0.6548	−0.0065
	**KAS**	10	4.941	3.284	-	0.7118	0.6762	−0.0557
**AER**	**KUT**	10	4.588	3.215	-	0.6941	0.6755	−0.0291
	**MUG**	19	5.824	3.457	1	0.6563	0.6409	0.0242
	**AYD**	6	4.529	3.209	1	0.7139	0.6863	0.0424
**CAR**	**KON**	15	5.471	3.399	-	0.6667	0.6830	0.0247
**MDR**	**ISP**	10	5.118	3.219	-	0.6610	0.6294	0.0503
	**KAH**	13	5.529	3.599	1	0.6886	0.6670	0.0325
	**ANT**	30	6.471	3.587	1	0.6887	0.6983	−0.0142
**EAR**	**KAR**	30	6.471	3.764	3	0.6919	0.6819	0.0148
**SAR**	**MAR**	40	6.647	3.731	1	0.6888	0.6791	0.0143
	**SAN**	31	6.412	3.803	-	0.6863	0.6831	0.0046

^1^ Marmara Region (MRM): Kırklareli—KIR, İstanbul/Çatalca—CAT, Malkara—MAL; Black Sea Region (BSR): Amasya/Merzifon—MER, Tokat—TOK, Kastamonu—KAS; Aegean Region (AER): Kütahya—KUT, Muğla—MUG, Aydın—AYD; Central Anatolian Region (CAR): Konya—KON; Mediterranean Region (MDR): Isparta—ISP, Kahramanmaraş—KAH, Antalya—ANT; Eastern Anatolia Region (EAR): Kars—KAR; South East Anatolian Region (SAR): Mardin—MAR, Şanlıurfa—SAN. *** *p* < 0.001

**Table 4 animals-10-01093-t004:** Genetic diversity parameters estimated for 17 microsatellite markers over all populations. TNA—total number of alleles; Na—mean number of alleles; Ne—effective number of alleles; PIC—polymorphic information content for each locus; F statistics (Fis, Fst, Fit); H_O_—observed heterozygosity; H_E_—expected heterozygosity; H_T_—Nei’s gene diversity; Hs—diversity within breeds; D_ST_—diversity between breeds; G_ST_—coefficient of gene differentiation; HWE—test for significant deviation from Hardy-Weinberg equilibrium with the hypothesis of the heterozygote excess; SR—Size range of the observed allele in bp.

Locus	TNA	Na	Ne	PIC	F_IS_	F_ST_	F_IT_	H_O_	H_E_	H_T_	Hs	D_ST_	G_ST_	HWE	SR	F(null)
HMS07	7	3.938	2.988	0.390	−0.0602	0.0201	−0.0389	0.449	0.438	0.447	0.436	0.011	0.025	NS	160–178	0.0341
ASB23	9	5.250	4.586	0.740	0.0169	0.0103	0.0270	0.737	0.757	0.763	0.758	0.005	0.006	NS	153–169	0.0106
HMS02	10	5.813	4.243	0.960	0.0058	0.0137	0.0195	0.716	0.708	0.716	0.708	0.007	0.010	NS	221–243	0.0087
COR058	11	6.438	4.921	0.760	0.0239	0.0128	0.0364	0.747	0.762	0.778	0.763	0.016	0.020	NS	187–209	0.0143
HMS03	8	4.438	3.276	0.540	0.0538	0.0034	0.0570	0.579	0.615	0.623	0.616	0.007	0.011	NS	149–169	0.0525
VHL209	9	4.313	3.228	0.450	−0.0100	0.0140	0.0041	0.473	0.467	0.476	0.466	0.010	0.021	NS	76–92	0.0000
ASB02	4	2.500	2.154	0.360	−0.0241	0.0008	−0.0232	0.443	0.445	0.445	0.446	−0.001	−0.002	NS	157–163	0.0000
HMS20	7	5.500	3.794	0.600	−0.0028	0.0000	−0.0032	0.666	0.663	0.665	0.663	0.001	0.002	NS	115–131	0.0097
COR007	7	5.375	4.588	0.710	0.0123	0.0124	0.0246	0.717	0.717	0.734	0.718	0.017	0.023	NS	165–177	0.0144
HTG07	11	8.813	6.164	0.810	0.0211	0.0033	0.0244	0.802	0.827	0.833	0.828	0.005	0.006	NS	136–158	0.0601
HTG10	9	7.250	5.321	0.480	−0.0041	0.0078	0.0038	0.796	0.787	0.799	0.787	0.012	0.015	NS	84–104	0.0583
COR018	11	7.188	5.625	0.800	−0.0198	0.0124	−0.0072	0.835	0.804	0.814	0.803	0.011	0.014	NS	252–276	0.0044
COR071	8	6.500	4.997	0.970	0.0125	0.0096	0.0220	0.756	0.755	0.765	0.755	0.010	0.013	NS	193–207	0.0361
COR082	6	4.563	3.955	0.650	0.2959	0.1955	0.4336	0.418	0.582	0.701	0.587	0.114	0.162	NS	214–224	0.1708
HTG06	4	3.938	3.741	0.660	−0.0007	0.0081	0.0073	0.721	0.711	0.716	0.711	0.005	0.007	NS	78–84	0.1894
LEX54	9	5.688	4.234	0.970	−0.0003	0.0000	−0.0031	0.679	0.675	0.673	0.675	−0.002	−0.004	NS	168–192	0.0278
AHT05	12	9.625	6.966	0.980	0.0167	0.0047	0.0213	0.857	0.884	0.890	0.885	0.004	0.005	NS	130–158	0.0264
All loci	142	5.714	-	0.696	0.0194	0.0192	0.0382	0.658	0.670	0.696	0.683	0.014	0.020			

**Table 5 animals-10-01093-t005:** Hierarchical AMOVA analysis among the 16 Turkish donkey populations.

Source of Variation	Variance Component (Estimate)	Variance (%)	Fixation Index	*p*-Value ^a^
Hypothesis 1: Geographical distribution				
Among groups	0.06303 (V_a_)	1.07	Φ_IS_: 0.01728	0.0000 ***
Among populations within groups	0.05650 (V_b_)	0.96	Φ_SC_: 0.00966	0.0254 *
Among individuals within populations	0.10009 (V_c_)	1.69	Φ_CT_: 0.01066	0.0000 ***
Within individuals	5.69268 (V_d_)	96.29	Φ_IT_: 0.03714	0.0137 *
Hypothesis 2: Reynold’s genetic distances distribution				
Among groups	0.12828 (V_a_)	2.16	Φ_IS_: 0.01728	0.0000 ***
Among populations within groups	0.02932 (V_b_)	0.49	Φ_SC_: 0.00504	0.0284 *
Among individuals within populations	0.10009 (V_c_)	1.68	Φ_CT_: 0.02156	0.0000 ***
Within individuals	5.69268 (V_d_)	95.67	Φ_IT_: 0.04331	0.0000 ***

^a^: *** *p* < 0.001; * *p* < 0.05.

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
