# Peer review of "Genetic Characterization of Native Donkey (Equus asinus) Populations of Turkey Using Microsatellite Markers"

_animals, 2020, doi:10.3390/ani10061093_

Round 1

Reviewer 1 Report

The manuscript has the merit to investigate a species which is quite neglected, nevertheless it's importance for the phylogeny of ungulates as well as for its large employment in developing countries.

The sampling and the methods of analysis are adequate and the outcome could be a good reference for further studies.

The only concern I have regards the K best fitting which unsurprisingly the Authors find to be equal 2. I suggest the reading of a Molecular Ecology paper from Janes et al (2017) https://onlinelibrary.wiley.com/doi/full/10.1111/mec.14187

and in particular: "ΔK method was proposed to assist in the identification of the “true” number of clusters. In our review of 1,264 studies using structure to explore population subdivision, studies that used ΔK were more likely to identify K  =  2 (54%, 443/822) than studies that did not use ΔK (21%, 82/386). A troubling finding was that very few studies performed the hierarchical analysis recommended by the authors of both ΔK and structure to fully explore population subdivision." That said, I suggest the Authors to follow what I put in bold or otherwise discuss the issue.   A minor point is to clarify in Table4 legend the meaning of F(null).   The English is generally correct but it might be improved by a mother tongue for phrases that look quite uncommon to a "pure" English.  

Author Response

Response to reviewer’s comments

The authors would like to thank the reviewer for constructive comments and suggestions.

Reviewer 1

 The only concern I have regards the K best fitting which unsurprisingly the Authors find to be equal 2. I suggest the reading of a Molecular Ecology paper from Janes et al (2017)

https://onlinelibrary.wiley.com/doi/full/10.1111/mec.14187

and in particular: "ΔK method was proposed to assist in the identification of the “true” number of clusters. In our review of 1,264 studies using structure to explore population subdivision, studies that used ΔK were more likely to identify K  =  2 (54%, 443/822) than studies that did not use ΔK (21%, 82/386). A troubling finding was that very few studies performed the hierarchical analysis recommended by the authors of both ΔK and structure to fully explore population subdivision." That said, I suggest the Authors to follow what I put in bold or otherwise discuss the issue.  

Answer: First of all, the authors want to thank you for your article suggestion. We read the article. However, as stated in the article, we do not think that we have misinterpreted the data under the K = 2 trend. In our study, the results of 3 different analysis methods support each other (FST, Neighbour-net dendrogram results, Structure results). The results obtained in all 3 analysis methods showed that there are two lineages or clade. It was also shown that some populations are found between the 2 clades. So K=2 is the true number of clusters for our study supported also by FST and Neighbour-net dendrogram data.

  • A minor point is to clarify in Table4 legend the meaning of F(null).  

 Answer: Additional information about the null allele has been added in the revised text.

Reviewer 2 Report

Please find the attached document to see the suggestions for minor modifications.

Author Response

Response to reviewer’s comments

The authors would like to thank the reviewer for constructive comments and suggestions. All the suggested revisions were done in the main text.

Herewith I suggest the MS entitled ‘Genetic characterization of Native Donkey (Equus asinus) Populations of Turkey using microsatellite markers’ for publication after modifications suggested below.

  • ...I clade (KIR, CAT, KAR, MAR, SAN) and II clade (MAL, MER, TOK, KAS, KUT, 40 KON, ISP, ANT, MUG, AYD and KAH) populations

please change to

...clade I (KIR, CAT, KAR, MAR, SAN) and clade II (MAL, MER, TOK, KAS, KUT, 40 KON, ISP , ANT, MUG, AYD and KAH) populations

Answer: Changed

  • In the entire te xt please use clade I, clade II, etc, (not I clade, II clade,…)

Answer: Changed

  • … largest change in the log of the likelihood function (DK) was…

Please use deltaK (with greek letter) like in the text (see line 151).

Answer: DeltaK is used instead of DK

  • ...Turkey had been extinct without been identified clearly

please change to

.           ..Turkey had been extinct without clear identification

Answer: Changed

  • …in other unpublished records, Mardin White…

Please provide which organisation has these records.

Answer: It is not an organization; there is a book about Turkish donkey breeding written in Turkish. In this book some local Turkish donkey breeds were mentioned but unfortunately most of them are extinct now. We have added that citation, cite number 6.

  • …breeds are also reported to be found in Turkey.

please change to

…breeds are also found in Turkey.

Answer: Changed

  • …if the mentioned breeds are still…

please change to

…if the above mentioned breeds are still…

Answer: added

  • …in Turkey is about 150.000…

please change to

…in Turkey is about 150,000…

Answer: Changed

  • …the first applying of molecular markers to characterize the donkey breeds …

please change to

.. the first application of molecular markers to characterize the donkey breeds…

Answer: Changed

  • In table 1:

please delete dots in front of the number 6

Answer: Deleted

  • …lower than the value of HE, indicating an excess of homozygosity…

please change to

…lower than the value of HE, indicating an excess of homozygosity…

Answer: Changed

  • …Our FST values fall into a small range, 0.00 – 563 (varying…

probably the authors wanted to write

…Our FST values fall into a small range, 0.000 – 0.056 (varying…

Answer: Changed

  • Clustering analysis by structure for the full-loci dataset assuming K = 2 to 20.

please change to

Clustering analysis by structure for the full-loci dataset assuming K = 2.

Answer: Changed

  • …population size, Turkish donkey population genetic diversity seems still conserved.

please change to

…population size, the genetic diversity of Turkish donkey population seems still conserved

Answer: Changed

Reviewer 3 Report

This is a careful and extensive study of a species that is often overlooked and that needs to be a target of conservation efforts. The results and interpretation can clearly help to drive the conservation of this genetic resource.

The strengths of the paper are that it takes a broad cross-section of Turkish donkey populations and then does genetic comparisons. In that sense, it is an extensive and inclusive look at the populations. It is generally quite strong, although one potential weakness is the relatively low sample numbers of some populations. That likely has no solution, given the problems of accessing animals and then getting samples. The main claims are that Turkey has several distinct donkey populations, but that these sort themselves out into two major groups of genetic resources that are related within the group. This can greatly help effective conservation practices, especially in the light of declining population numbers. The paper is meaningful both for donkeys specifically (which is admittedly a small audience) but also for the way in which it addresses the conservation question of deciding what to include within a distinct population for genetic management. Their approach will therefore be more broadly useful, especially in developing countries or other areas with local breeds or landraces that are not organized into formal breeds with breeder associations and registries. The claims are convincing, and no further evidence is needed. They are discussed appropriately. Other than the relatively small sample size of a few populations I don't see many problems, and that problem is minor. The English is completely understandable, although obviously not written by a native speaker. That is a relatively minor drawback in this situation.

Author Response

The authors would like to thank the reviewer for constructive comments for our manuscript.